# In Vitro Effects of Paralytic Shellfish Toxins and Lytic Extracellular Compounds Produced by *Alexandrium* Strains on Hemocyte Integrity and Function in *Mytilus edulis*

**DOI:** 10.3390/toxins13080544

**Published:** 2021-08-05

**Authors:** Virginia Angélica Bianchi, Ulf Bickmeyer, Urban Tillmann, Bernd Krock, Annegret Müller, Doris Abele

**Affiliations:** 1Laboratorio de Ecotoxicología Acuática, INIBIOMA (CONICET-UNCo)—CEAN, Ruta Provincial N° 61, Km 3, CCP 7, Junín de los Andes, Neuquén 8371, Argentina; 2Alfred Wegener Institute for Polar and Maine Research, Am Handelshafen 12, 27570 Bremerhaven, Germany; ulf.bickmeyer@awi.de (U.B.); urban.tillmann@awi.de (U.T.); bernd.krock@awi.de (B.K.); annegret.mueller@awi.de (A.M.)

**Keywords:** neurotoxins, bioactive extracellular compounds, immune response, blue mussel

## Abstract

Harmful effects caused by the exposure to paralytic shellfish toxins (PSTs) and bioactive extracellular compounds (BECs) on bivalves are frequently difficult to attribute to one or the other compound group. We evaluate and compare the distinct effects of PSTs extracted from *Alexandrium catenella* (Alex5) cells and extracellular lytic compounds (LCs) produced by *A. tamarense* (NX-57-08) on *Mytilus edulis* hemocytes. We used a 4 h dose–response in vitro approach and analyzed how these effects correlate with those observed in a previous in vivo feeding assay. Both bioactive compounds caused moderated cell death (10–15%), being dose-dependent for PST-exposed hemocytes. PSTs stimulated phagocytic activity at low doses, with a moderate incidence in lysosomal damage (30–50%) at all tested doses. LCs caused a dose-dependent impairment of phagocytic activity (up to 80%) and damage to lysosomal membranes (up to 90%). PSTs and LCs suppressed cellular ROS production and scavenged H_2_O_2_ in in vitro assays. Neither PSTs nor LCs affected the mitochondrial membrane potential in hemocytes. In vitro effects of PST extracts on *M. edulis* hemocytes were consistent with our previous study on in vivo exposure to PST-producing algae, while for LCs, in vivo and in vitro results were not as consistent.

## 1. Introduction

Dinoflagellate species of the genus *Alexandrium* are responsible for harmful algal blooms (HAB) affecting marine ecosystems, with massive socio-ecological consequences resulting worldwide [1,2,3]. *Alexandrium* spp. can produce different types of toxic or harmful bioactive compounds, with reported deleterious effects on the aquatic fauna, including commercially interesting species of fish and bivalves [3]. In this context, HAB exposure has been related to altered immune functions and global physiological distress, producing increased susceptibility to diseases in bivalves [4]. An unambiguous attribution of the harmful effects to one or another compound group remains, however, difficult.

One important group of compounds produced by a number of *Alexandrium* species are paralytic shellfish toxins (PSTs). PSTs have been known for more than 70 years [5], are chemically well-characterized [6], and their mode of action as blockers of sodium ion channels in vertebrate nerve cells has been elucidated (reviewed in [7]). In contrast, little is known about their biological role. The PST group includes the highly neurotoxic saxitoxin (STX) as well as its variants such as neosaxitoxin (NEO), gonyautoxins1–4 (GTX1–GTX4), and the *N*-sulfocarbamoyl PST variants called B1/2 and C1–4 toxins [8]. All of these compounds are not readily released into seawater by intact cells [9]; instead, *Alexandrium* cells are consumed by algal grazers such as filter-feeding bivalves, and toxins are released from the algae into the animal’s digestive tract. Even though invertebrates are much more tolerant to phycotoxins than vertebrates, including marine mammals and humans, exposure to a HAB of *Alexandrium* spp. may still produce neurotoxic and/or cytotoxic effects in bivalves, which cause tissue damage, paralysis, altered behavior, loss of coping strategies, and mass mortalities (reviewed in [3]). During the feeding process, the immune defense system, gut microbiota, and physiological mechanisms of toxin biotransformation allow bivalves to metabolize, accumulate, and/or eliminate PSTs, e.g., [10,11]. Part of the defense against toxic algae involves the hemolymphatic cellular immune response, also involved in xenobiotic metabolism and tissue regeneration processes [12,13]. Administration of PST-producing algae, pure single PST, or a mix of these toxins can also trigger immunological responses in clams [9,14,15], oysters [16,17], and mussels [18]. However, whether PSTs produce immunotoxic or immunostimulant effects on bivalves depends on the species’ resistance, their toxin exposure history, and the ingested doses. In particular, *Mytilus* spp. display a high level of immunological vigor toward adverse environmental conditions compared to other marine bivalves [19]. Galimany et al. [20] suggest that PSTs spread throughout the hemocyte cytoplasm and accumulate in lipofuscin granules in order to be eliminated in the intestinal lumen. In addition, immunological modulation involving hemocyte mobilization, phagocytic activity, reactive oxygen species (ROS) production, antioxidant responses, and reduced cellular viability were observed in PST-exposed mussels [18,20,21].

Many species of *Alexandrium*, including many of the PST-producing ones, also produce bioactive extracellular compounds (BECs), which are clearly unrelated to the known PSTs [22]. In contrast to PSTs, BECs are released to the surrounding aquatic medium [9,23], where they inhibit the growth and survival of competing algae and protistan grazers [24,25]. In contrast to their well-documented biological functions and ecological effect, the chemical nature of BECs is poorly characterized. Particularly, the production of BECs with lytic activity (Lytic compounds, LCs) was shown for *Alexandrium*. These LCs are large amphipathic compounds (between 7 and 15 kDa) that do not belong to any group of known biopolymers such as peptides or polysaccharides [26]. Their lytic activity is stable over wide ranges of temperatures and pH and are refractory to bacterial degradation, but these compounds generally show low recovery in common purification techniques [27].

Growing evidence documents the effect of BECs on the immunological functions of bivalves exposed to LC-producing algae, e.g., [28,29,30,31]. Particularly, the mechanisms of action of LCs include the disturbance of cell membrane integrity, with some indications that contact between LCs and certain sterol molecules in the membranes are involved [23]. In vitro immunotoxic effects of *Alexandrium* strains that do not produce PSTs were evidenced to diminish phagocytic activity and surface cellular adherence in hemocytes of two clam species, which was attributed to the hypothetical action of LCs on cellular osmotic balance [8]. It has further been suggested that the direct cytotoxic effect of LCs on hemocytes occurs when: (i) hemocytes infiltrate the gill tissue and intestinal lumen of the bivalves and get in contact with LCs and/or producer algae, and (ii) when hemocytes get in contact with LCs containing seawater that infiltrates the open circulatory system of the bivalves [9,29,30].

Whereas numerous studies on the physiological effects of PSTs in bivalves exist, similar studies dealing with BECs are scarce. However, in both cases, studies are mostly focused on in vivo assays, while in vitro assays are scant. Furthermore, many studies investigating the effects of PSTs use *Alexandrium* cultures in which the possible effect of BECs is not evaluated, and thus cannot differentiate between the effects of both compound classes, and/or attribute the effects only to PSTs. In Bianchi et al. [31], we showed that feeding *Mytilus edulis* with PST and/or LC-producing *Alexandrium* strains (Alex5 and Alex NX-57-08, respectively) caused similar rates of hemocyte damage and immune function impairment after three days. These effects were mostly transitory, vanishing within a seven-day experimental period, with the exception of lysosomal membrane damage that accrued when PSTs and LCs were combined. However, PST amounts and LC activity differed among algal strains used in this in vivo exposure, which implies some difficulty in interpreting and comparing individual and combined effects. Therefore, additional in vitro experiments were conducted to evaluate how hematological effects in *M. edulis* relate to the action of each algal toxin type on the hemocytes, and how these effects correlate with those observed in vivo. Increased knowledge of the physiological responses and deleterious effects of bioactive compounds from dinoflagellates on bivalves, integrating in vivo and in vitro approaches, may help to explain their mechanisms of action and predict further consequences in the field. The aim of this work was to evaluate and compare the distinct effects of PST extracts and lytic culture supernatant (extracellular LCs) produced by *Alexandrium* strains (Alex5 and Alex NX-57-08, respectively) on hemocyte integrity and immune function, using a dose–response in vitro approach. In addition, a comparative assessment of present in vitro and previously published in vivo results is discussed.

## 2. Results

### 2.1. PST and LC Effects on Hemocytes

#### 2.1.1. Cytotoxicity

No significant effect on the cellular count was detected after 4 h of treatment with PST or LC doses (ANOVA *p* = 0.36 and *p* = 0.18, respectively). Exposure of *M. edulis* hemocytes to extracted PSTs elicited a dose-dependent effect on cell viability (exponential function: Viability = 93.74 e^(−0.0019 x)^, S = 1.98; EC_50_ 67.54 µM PST, corresponding to 373 *A. catenella* cells mL^−1^; CI 95%: 192 to 852), with a maximal diminution of 15% viability. Viability was significantly affected (ANOVA *p* ˂ 0.01) at 125–500 µM (corresponding to 690–2760 *A. catenella* cells mL^−1^) (Dunnett’s test *p* ˂ 0.05), with an additional effect at the highest concentration (1250 µM, corresponding to 6900 *A. catenella* cells mL^−1^, Dunnett’s test *p* ˂ 0.01) compared to control (Figure 1a). Exposure to LCs also resulted in 10–15% loss in cell viability when hemocytes were exposed to culture supernatant from 340 to 6800 *A. tamarense* cells mL^−1^ compared to control (ANOVA *p* ˂ 0.001; Dunnett’s test *p* ˂ 0.01 for 340 and 1700 cells mL^−1^; *p* ˂ 0.001 for 680 cells mL^−1^; *p* ˂ 0.05 for 3400 and 6800 cells mL^−1^) (Figure 1b). Lysosomal membrane stability decreased by 30–50% compared to control level upon exposure to all PST concentrations (ANOVA *p* ˂ 0.01; Dunnett’s test *p* ˂ 0.05) (Figure 1c). For LCs, this marker decreased in a dose-dependent manner (exponential function: Lysosomal membrane stability = 0.019 e^(−0.0025 x)^, S = 0.002; EC_50_ 274 *A. tamarense* cells mL^−1^; CI 95%: 153–1345) and by almost 90% in cells exposed to the exudate from the highest algal cell densities (ANOVA *p* ˂ 0.05; Dunnett’s test *p* ˂ 0.5 for 170 cells mL^−1^; *p* ˂ 0.01 for 340 and 680 cells mL^−1^; *p* ˂ 0.001 for 1700–6800 cells mL^−1^, compared to control) (Figure 1d).

#### 2.1.2. Cellular Function

Phagocytic activity changed with increasing PST concentration to a maximum in the 62.5 and 125 µM treatments corresponding to 345 and 690 *A. catenella* cells mL^−1^ (ANOVA *p* ˂ 0.001; Dunnett’s test *p* ˂ 0.01 for both), without further changes at higher concentrations with respect to control (Figure 2a). LCs impaired phagocytosis in a dose-dependent manner (exponential function: Phagocytic activity = 0.64 e ^(−0.0033 x)^, S = 0.07; EC_50_ 211 *A. tamarense* cells mL^−1^; CI 95%: 125–683). At the highest doses, phagocytic activity was 80% lower than in controls (ANOVA *p* ˂ 0.0001; Dunnett’s test *p* ˂ 0.05 for 170 cells mL^−1^; *p* ˂ 0.01 for 340 and 680 cells mL^−1^; *p* ˂ 0.001 for 1700–6800 cells mL^−1^) (Figure 2b).

Hemocyte extracellular H_2_O_2_ production decreased by about 60% in all PST treatments compared to control (ANOVA *p* ˂ 0.001; Dunnett’s test *p* ˂ 0.001, for all comparisons) (Figure 3a). The effect of LC containing the supernatant on hemocytes diminished extracellular H_2_O_2_ levels only for 680 and 3400 *A. tamarense* cells mL^−1^ (ANOVA *p* ˂ 0.05; Dunnett’s test *p* ˂ 0.01 and *p* ˂ 0.05, respectively) (Figure 3b). Intracellular ROS formation was affected only at the highest exposure concentration and diminished by 30% for both PST and LC exposure compared to control (ANOVA *p* ˂ 0.05; Dunnett’s test *p* ˂ 0.05, for both comparisons) (Figure 3c,d).

Mitochondrial membrane potential (MMP) was examined only in hemocytes attached to the coverslip that show no conspicuous change in morphology, following 4 h of incubation. MMP in hemocytes was neither affected by PST (1250 µM, corresponding to 6900 *A. catenella* cells mL^−1^) nor LC (supernatant from 6800 *A. tamarense* cells mL^−1^) exposure (Figure 4). Immune cell clumping was only seen in the PST treatment.

Cytotoxic and functional variables tested in hemocytes in the dose–response in vitro approach were correlated with each other and presented as a Pearson correlation matrix (Table 1). For PST exposure, variation in cellular viability showed a strong positive correlation with lysosomal membrane damage; however, all these variables had no relationship with phagocytic activity. For LC, the lysosomal membrane stability variation showed a significantly positive correlation with cellular viability and phagocytosis. For both bioactive compound classes, there was no relationship between ROS levels and phagocytosis. The correlation between cellular viability and ROS levels is not further discussed since ROS was measured in living cells, so these variables are not actually related.

### 2.2. Toxin Scavenger Function

#### 2.2.1. In Vitro Chemical Effects of PST and LC on H_2_O_2_ Levels

Both PSTs and LCs dampened H_2_O_2_ concentration in vitro. PSTs significantly reduced H_2_O_2_ levels at all tested concentrations as compared to control (ANOVA *p* ˂ 0.01; Dunnett’s test *p* < 0.05 for 250 and 500 µM, corresponding to 1380 and 2760 *A. catenella* cells mL^−1^; *p* < 0.01 for 1250 µM, corresponding to 6900 cells mL^−1^) (Figure 5a). Addition of LCs to the H_2_O_2_ detecting system caused a dose-dependent scavenging effect across all concentrations (linear regression, r^2^ = 0.77, *p* < 0.001), with a significant effect on the supernatant from 6800 *A. tamarense* cells mL^−1^ (ANOVA *p* ˂ 0.01; Dunnett’s test *p* < 0.05) (Figure 5b).

## 3. Discussion

From our in vitro dose–response studies, we can show and compare the separate actions of different dinoflagellate secondary metabolites on the immune cells of the blue mussel *Mytilus edulis*. PSTs and LCs from *Alexandrium* strains (*A. catenella* strain Alex5 and *A. tamarense* strain Alex NX-57-08, respectively) are bioactive substances that affect the cellular integrity and immune function of hemocytes, acting dose dependently in some cases. However, interpretation of functional and cytotoxic in vitro markers has to be performed with caution when comparing different published results. Generally, it has to be considered that PST extracts and LC containing supernatant are not pure or single compounds. Using dilute acetic acid as an extraction solvent for PSTs, mainly inorganic and polar organic compounds are extracted, and the majority of organic cell contents are excluded, but the presence of other molecules in PST extracts cannot be discarded. Likewise, the culture supernatant may include BECs other than the lytic compounds, and lytic bacteria may also be present, as algal cultures were not axenic. Further systematic testing including additional controls (i.e., the cellular extract of a non-PST strain; supernatant of a non-BEC strain), and/or with pure PSTs (single compounds and mixtures) would complement these assessments.

### 3.1. PST vs. LC: In Vitro Cytotoxic and Functional Effects

The exponential dose-dependent increase in cellular death, observed in our in vitro assay for *M. edulis* hemocytes exposed to *A. catenella* (Alex5) PST extract, is likely due to caspase-dependent apoptosis, as has been shown for hemocytes of many bivalve species exposed to STX, gonyautoxins (e.g., GTX2/3), or B1 and C1/C2 [32,33]. Indeed, Abi-Khalil et al. [33] found that a few hours of exposure to elevated concentrations of purified PSTs cause dose-dependent morphological alterations, the permeabilization of cytoplasmic and nuclear membranes, DNA damage, and hemocyte death in oysters. In our study, cellular death is positively correlated with the loss of lysosomal membrane stability in *M. edulis* hemocytes exposed to PST extract. This subcellular damage could be explained based on previous studies by (i) a release of intracellular lytic enzymes, triggered as part of the detoxification process against PSTs [20], but not directly correlated with phagocytosis in our work; and (ii) by a direct binding of toxin molecules to membrane ion channels affecting the lysosomal membrane functionality [34]. In fact, the regulation of cell membrane permeability, by blocking Na^+^ channels under ionic stress conditions, is one of the biological functions suggested for STX in the producer algae (reviewed by [35]). The cytotoxic effect of LCs is not related to the induction of programmed cell death but to the direct disruption of cytoplasmic membranes causing their permeabilization and altering the cellular homeostatic balance [23]. In our work, the lethal effect on *M. edulis* hemocytes is restricted and stabilizes as LC concentrations increase. This limited sensitivity of membranes to the lysis could be showing the variable content of targeted constituents such as sterols and phosphatidylcholine [26] expressed in the hemocyte subpopulations [36].

All in all, the cytotoxicity parameters evaluated for PST exposure on *M. edulis* hemocytes appears to be of limited severity in our study, which could be attributed to known mechanisms of ligand–receptor interaction for toxin elimination [20], cellular ion homeostasis [21], and regeneration capacity [18]. Contrary to PSTs, there is no evidence of LCs entering the cytoplasm of live cells, but the resulting harmful effect at the subcellular level is indeed stronger for LC-containing supernatant than for PST-containing cell extract. This subcellular effect is positively correlated with cellular death and is displayed as an exponential function of the dose-dependent labilization of lysosomal membranes, probably linked to a general homeostatic destabilization in hemocytes such as changes in cytoplasmic pH and osmolarity [34].

The cellular immune response is distinctly affected by the PST extract and extracellular LCs. Despite increased cellular death and lysosomal destabilization caused by PST exposure in *M. edulis* hemocytes, there is no correlation between these individual variables and phagocytic activity for yeast cells, which is indeed stimulated at low doses. These toxins can be recognized by hemocytes as non-self molecules as they trigger the over-expression of Pattern Recognition Receptor (PRR) genes related to the recognition of glycan and carbohydrates (such as galectins and lectins) in fungi cell walls [18,21,37]. An overwhelmed receptor response in *M. edulis* hemocytes exposed to increased toxin dosage could explain the decrease of phagocytosis to control levels, affected by the PST extract (Figure 2a). ROS production does not correlate with the phagocytic activity induced by PSTs in *M. edulis* hemocytes. In fact, these toxins do not elicit much intracellular ROS production and diminish the amounts of ROS released into the medium. This was corroborated in our in vitro H_2_O_2_ scavenging test, even at the lowest toxin concentration (250 μM, corresponding to 1380 *A. catenella* cells mL^−1^). PST antioxidant potential could account for a biological function neutralizing excessive ROS production in algae cells such as when these microorganisms are under oxidative stress conditions during a “bloom” [38]. PST effect on hemocyte intracellular ROS production at the highest tested doses seems to be more of an indirect effect and not related to the disruption of the MMP, which is unaffected by the toxin in our in vitro assay. The increased expression of mitochondrial enzymes and voltage-gated channel proteins was observed in *M. chilensis* following STX exposure, which is suggested as a mechanism for maintaining mitochondrial stability [21]. Contrary to PSTs, subcellular destabilization caused by the exposure to LC containing supernatant is strongly correlated with phagocytic activity, which decreases at a dose-dependent exponential function, probably because osmotic unbalance also impairs invagination and adhesion processes [9]. The LC effect on cellular ROS production is not clear, except at a very high concentration where intracellular ROS levels decrease. This is not accompanied by a change in MMP in *M. edulis* hemocytes at the same LC concentration. Indeed, the H_2_O_2_ scavenging test revealed a linear relationship between quenching and the applied LC doses, which evidences a direct and clear antioxidant effect of the compounds from the *Alexandrium* culture supernatant. It has been speculated that extracellular ROS is involved in the toxicity of *Alexandrium* and other HAB species [39], but this is not necessarily due to a direct effect on target organisms. Instead, synergistic interactions between ROS and PUFA/LC-PUFA released by *Alexandrium* may lead to the production of toxic secondary metabolites by lipid peroxidation processes [40]. It has been shown that scavenging of extracellular ROS impairs growth population in HAB species [41,42]. Although, the possible allelochemical/physiological role of the antioxidant effect of BECs from *A. tamarense* (NX-57-08) evidenced in our results has not been elucidated. Increased knowledge of the chemical nature of BECs (lytic and others) is necessary to fully describe this process.

### 3.2. In Vitro vs. In Vivo Effects

Few publications have addressed and compared the in vivo and in vitro effects of bioactive compounds in bivalve model species, which limits our ability to compare the present results and draw definite conclusions toward the biochemical mechanisms underlying their mode of action. In a previous study from this project, mussels were fed the same *Alexandrium* strains used in the present work, Alex5 (PST) and Alex NX-57-08 (LC), for three and seven days, in order to study the in vivo effects of both compound types and those of their combined application (also using a PST+LC producer strain). PST and LC doses used for in vitro exposure correspond to those produced by *Alexandrium* spp. at a density that *M. edulis* can easily consume/tolerate during in vivo exposure [31]. Accordingly, these algae densities correspond to environmentally realistic conditions registered for HABs of *Alexandrium*, which are relevant to their effects on bivalves, e.g., [43,44]. Finally, mussels used for in vitro and in vivo exposure assays were collected in the same season and at the same location, were of the same size, and maintained under the same acclimatization conditions. In what follows, we will mainly compare the results of the in vivo and in vitro approaches in our study, individually for each substance class.

Our results indicate that the in vivo and in vitro effects of PST on *M. edulis* hemocytes are consistent and follow coherent patterns. Cellular viability is similarly affected when mussels are fed for three and seven days with PST-containing but non-lytic *A. catenella* strain Alex5 (10–20% loss of viability) cells, as when hemocytes are directly exposed to PST extract over 4 h (max. 15% viability loss). Dose-dependent effects on cellular death rates are corroborated in vitro from 31.25 to 1250 µM (172–6900 cells mL ^−1^). In addition, labilization of lysosomal membranes is evident and amounts to ~40% damaged cells in vivo, and between 30 and 50% in vitro. Phagocytic activity is stimulated in vivo and in vitro; in both cases, the effect is limited and returns to control level at higher PST concentrations, which, for the in vivo experiment, would emulate the accumulation of ingested toxin after seven days. A strong but transient scavenging effect of PSTs on hemocyte ROS production is observed after three feeding days (60% for extracellular—80% for intracellular) with return to control levels after seven days. This antioxidant effect of the PST extract is corroborated by in vitro tests, with hemocytes exposed to the highest PST dosage, and additionally confirmed in the direct biochemical assays of the H_2_O_2_ scavenging capacity of the samples. These results suggest that cytotoxic effects and initially diminished ROS levels would not impair immunological competence against non-self particles in *M. edulis*, as in vivo effects are corroborated by in vitro approaches. There are several studies dealing with the effects of STX in vitro exposure and simulated blooms of *A. minimum* on hemocyte viability, phagocytic activity, and ROS production in hemocytes of *C. gigas* [4]. Contrary to our findings, the cited investigations show inconsistent results which are attributed to the lack of homogeneity in bivalve populations and algae culture conditions used in these studies. Moreover, comparing single toxin effects with the exposures to living algae limits the accuracy of emerging conclusions.

Results obtained from in vivo and in vitro exposures to LCs are not as consistent, but the immunotoxic potential of LCs from *Alexandrium* is clearly evident in *M. edulis*. Cytotoxicity is similar for both exposure conditions, with hemocyte viability 10–20% affected in vivo and 10–15% in vitro. In addition, labilization of lysosomal membranes is observed in mussels fed with lytic *A. tamarense* strain NX-57-08 for three and seven days, with a strong dose-dependent effect corroborated in in vitro experiments. An impairment of phagocytic activity is only observed when LCs are directly administered to hemocytes in vitro but are absent when mussels are fed with producer algae. Similar to PSTs, LCs diminish intracellular and extracellular H_2_O_2_ levels in vivo only in the first experimental phase after three days of feeding. An antioxidant capacity of LC containing supernatant was corroborated in vitro in hemocytes exposed to the highest concentration and in the H_2_O_2_ scavenging assays. In vitro exposure allows for the direct contact of LCs with the hemocytes, while in vivo feeding exposure to the lytic strain may cause indirect deleterious effects driven by a global physiological stress derived from tissue damage, as previously seen in oysters and scallops [28,29,30]; these consequences may also include the impairment of hemocyte functions other than immunity (tissue repair, nutrition, biomineralization, etc.), as previously suggested for many bivalve species [4]. The difference between these exposure approaches would explain the inconsistency among cytotoxic and immunotoxic effects observed in *M. edulis* hemocytes under both exposure conditions. Consequently, these results highlight the difficulty in extrapolating mechanistic conclusions from in vitro and in vivo conditions for the exposure of mussels to LCs. A similar outcome is found when the in vitro exposure of *P. philippinarum* hemocytes to the *Karenia selliformis* cell-free culture supernatant caused cell death and the inhibition of ROS production [45], while in vivo exposure to a simulated bloom showed no detectable effect [46].

## 4. Conclusions

Bioactive compounds produced by *A. catenella* (Alex5) and *A. tamarense* (Alex NX-57-08) affect the integrity and functionality of *M. edulis* hemocytes exposed in vitro. Both compound classes induced similar levels of cell mortality. However, PSTs at low doses may act as immunostimulants of phagocytic activity despite causing moderate damage to lysosomal membranes; on the other hand, LCs are harmful and immunotoxic, impairing lysosomal stability and phagocytosis in a dose-dependent manner. Both compound classes elicit the suppression of intracellular ROS levels, which is not related to a disturbed mitochondrial membrane potential, but rather to an indirect physiological disturbance at high doses. Both PSTs obtained from cellular extracts and supernatant containing LCs have a direct scavenger action on ROS in vitro. However, the biological role of the antioxidant properties of these bioactive compounds has not been fully described thus far.

As discussed above, PST and LC doses used in the present study correspond to environmentally relevant algal densities, which bivalves may be exposed to during a HAB. After a comparative analysis of present and previously published results, we conclude that the extrapolation of in vitro cellular effects and responses to in vivo systems is more straightforward for PSTs than for LCs. These kinds of in vitro/in vivo combined approaches could be useful for the evaluation and validation of biomarker responses in the field, but also for the prediction of potential immunomodulatory effects on bivalves upon exposure to algae bioactive products.

## 5. Materials and Methods

### 5.1. Preparation of PST Extracts and Extracellular LCs

PSTs and LCs were obtained from cultures of *Alexandrium* strains, previously characterized through morphological and phylogenetical analyses. The clone Alex5 of *Alexandrium catenella* (formerly group I of the *A. tamarense*/*fundyense*/*catenella* species complex) isolated from the North Sea coast off Scotland [47] has been shown to produce PSTs but not LCs; however, a strain of *A. tamarense* (NX-57-08, formerly group III of the *A. tamarense*/*fundyense*/*catenella* species complex) isolated in 2015 from Trondheimfjord (Norway) (Tillmann, unpublished) produces LCs but not PSTs [31]. Both strains were grown in 250 mL glass Erlenmeyer’s flasks, with seawater K-medium supplemented with selenite, prepared from 0.2 mm sterile-filtered (VacuCap, Pall LifeSciences, Dreieich, Germany) North Sea seawater (salinity 32) at 15 °C, under cool-white fluorescent light at a photon flux density (PFD) of 100 μmol photons m^−2^ s^−1^ in a 16 h light:8 h dark photo-cycle. Cultures were harvested in the early stationary phase.

For PST extraction, cell pellets of *A. catenella*, obtained by centrifugation at 3200× *g* for 15 min, were transferred with a 500 µL extraction solvent (0.03 *N* acetic acid) into a FastPrep tube containing ca. 0.9 g lysing matrix D. The tubes were placed in a FastPrep instrument to homogenize the samples at a speed of 6.5 for 45 s. After that, samples were centrifuged for 15 min at 8100× *g* and 4 °C. The supernatant (400 µL) was filtered through a pore-sized 0.45 µm filter by centrifugation for 30 s at 800× *g*. Filtrate was transferred to an HPLC vial. Using acetic acid as an extraction solvent, mainly inorganic and polar organic compounds were extracted, and the majority of organic cell contents were excluded. Toxin profiles and content were analyzed by reverse-phase ion-pair liquid chromatography coupled to post-column derivatization and fluorescence detection (LC-FD) based on previously published methods [48,49]. Data were calibrated against external PST calibration curves prepared from standards that were purchased from the Certified Reference Material Programme of the Institute of Marine Biosciences, National Research Council, Halifax, NS, Canada. Results are expressed as ng per µL and µM of extract (Table 2). This PST solution was preserved at −20 °C for one month, before being used in the in vitro exposures.

For the assessment of the lytic capacity of bioactive compounds from *A. tamarense* (NX-56–08), the original culture containing 13,637 cells mL^−1^ was centrifuged (15 min at 3200× *g*), and the supernatant containing lytic bioactive compounds was slowly pipetted to avoid cell debris resuspension. The lytic capacity of the culture supernatant was quantified by a cryptophyte *Rhodomonas salina* (strain KAC30) bioassay according to Tillmann et al. [24]. EC_50_ was calculated by fitting data points (log-transformed *A. tamarense* cells mL^−1^, corresponding to the amount of supernatant in the sample, vs. % of *R. salina* intact cells) to a sigmoidal curve (see Tillmann et al. [25] for the sigmoidal curve equation) by non-linear regression using Statistica. The EC_50_ for this culture was 556 cells *R. salina* per mL with a 95% confidence interval of 518–597 (Figure 6). The supernatant was stored in the dark for one month at −20 °C before being used in the in vitro exposures, and such storage conditions have been reported to preserve lytic activity. Since LCs are known to attach to surfaces such as those of microcentrifuge tubes, a partial decrease in the supernatant lytic activity during hemocyte in vitro exposure cannot be discarded (Section 5.4) [27].

### 5.2. Mussel Collection and Maintenance

As previously published [31], adult *M. edulis* (44.9 ± 2.2 mm of shell length, commercial size) were collected on the island of Sylt and transported to the Alfred Wegener Institute in Bremerhaven. According to official controls, these areas are not affected by toxic blooms. Mechanically cleaned mussels were acclimatized for 21 days in tanks with circulating seawater (salinity 32.8, 8.13 ± 0.5 °C, 0.15 mg L^−1^ NH_4_^+^, 0.02 mg L^−1^ NO_2_^−^, 26.53 mg L^−1^ NO_3_^−^, pH 8.02, 10.68 mg L^−1^ dissolved oxygen). Mussels were fed every seven days with *Nannochloropsis salina* cells (PhytoMaxx N° 12009, Nyos^®^ Aquatics GmbH, Korntal-Münchingen, Germany) at manufacturer’s recommended amounts according to mussel number and water tank volume. No mortality was observed under these conditions during the acclimatization period.

### 5.3. Hemocyte Extraction and Manipulation

Mussels were kept on ice for 5 min before hemolymph extraction. Hemolymph was drawn from the posterior adductor muscle of each mussel using a sterile syringe, aliquoted in microcentrifuge tubes, and kept on ice for immediate analysis. The syringe was prefilled with a sterile Alseve medium (20.8 g L^−1^ glucose; 8 g L^−1^ sodium citrate; 3.36 g L^−1^ EDTA; 22.5 g L^−1^ NaCl; pH 7; 920 mOsm; [50]) (1:5, medium:hemolymph) as anticoagulant and a nutritive medium for the cells (except for phagocytic activity assays). Briefly, after extraction and before in vitro exposure, the total number (cells mL^−1^) and viability of hemocytes in each sample was determined microscopically in quadruplicate using a Fuchs Rosenthal chamber (see Section 5.4.2). Cellular density was adjusted using an Alseve medium, when appropriate.

### 5.4. PST and LC Effects on Hemocytes

#### 5.4.1. In Vitro Experimental Design

Hemocyte exposures to PSTs or extracellular LCs were carried out in microcentrifuge tubes kept in the dark, open and cold (over ice, 8–10 °C), with orbital shaking (Schüttler KS130 Control, IKA Werke, Staufen, Germany) at 0.05× *g*, for 4 h. Cellular viability under control in vitro conditions was checked in previous assays using the trypan blue staining method (see Section 5.4.2). Hemocyte viability was stable, at least up to 5 h (96.3 ± 0.6% at 1 h; 93.0 ± 1.0% at 2 h; 94.0 ± 4.0% at 3 h; 92.7 ± 1.2% at 4 h; 93.7 ± 2.1% at 5 h; one-way ANOVA *p* = 0.31, *n* = 6). Filtered sea water (FSW) used for toxin dilution was prepared in optimal physiological conditions previously measured in hemolymph: 920 mOsm (Vapro Osmometer 5500, Wescor Inc., Logan, UT, USA), salinity 25.4, and pH 7.4 (Microprocessor pH-mV Meter pH526, WTW, Weilheim, Germany). Filtered seawater in these new conditions will be called modified filtered seawater (mFSW) hereafter.

For PST exposure, an aliquot of the cell extract (see Section 5.1) was thawed and the acetic acid extract was dried under a stream of nitrogen. The residue was then combined in mFSW to reach 62.5, 125, 250, 500, 1000, and 2500 µM (working solutions). For extracellular LC exposure, an aliquot of the frozen supernatant was thawed, vigorously shaken, and diluted with deionized water (0.28 mL per mL of supernatant) to reduce salinity to 25.4. Working solutions were prepared by diluting this preparation with mFSW at 2.5, 5, 10, 25, 50, 100%. Both PST and LC working solutions were kept on ice and immediately used. Final concentrations were obtained from the in vitro assays by a 1:2 dilution (PST: 31.25, 62.5, 125, 250, 500, 1250 µM; LC: 1.25, 2.5, 5, 12.5, 25, 50%). Both PST and LC doses were chosen within the range of toxin concentrations that *M. edulis* could easily consume/tolerate during in vivo exposure [31]. Particularly for LCs, the concentration range corresponds to the densities of *A. tamarense* cells used in the *R. salina* assay (Section 5.1). For a comparative analysis of their physiological effects on hemocytes, PST/LC concentrations are expressed from now on as the number of producer *Alexandrium* cells mL^−1^ determined in the original culture and corresponding to the working concentrations used for the in vitro exposures. Doses and corresponding cell densities are listed in Table 3, from cultures of 4734 cell mL^−1^ of *A. catenella* (Alex5; 72.4 pg PST cell^−1^) and 13,600 cell mL^−1^ of *A. tamarense* (Alex NX-57-08).

Hemocyte in vitro exposures were carried out separately for each toxin and for each cellular variable following a repeated measures design, where each mussel hemocyte sample was aliquoted and exposed to PST or LC serial dilutions. Control treatments with mFSW were set for each sample and measurement.

#### 5.4.2. Cytotoxicity

Lysosomal membrane stability was assessed using the Neutral Red Retention (NRR) assay adapted for microplate reading by Coles et al. [51] and modified in Bianchi et al. [31]. For each toxin concentration, 300 μL hemolymph (233,500 ± 42,650 cells mL^−1^) was mixed with 300 µL of the corresponding PST or LC treatment and incubated as described in Section 5.4.1. (*n* = 4 by duplicate, each n is a hemolymph sample, pooled from two animals). A total of 300 μL of the 200 μM neutral red solution was subsequently added to the samples, incubated for 1 h and centrifuged (800× *g*, 5 min at 10 °C). A neutral red stain was extracted from the cell pellet with a lysis buffer (1% acetic acid and 50% ethanol in distilled water) and centrifuged again. Absorbance of the supernatant was read at 550 nm in a 96-well plate using a microplate reader (TRISTAR, Fa. Berthold, Bad Wildbad, Germany). Results are expressed as optical density (OD) per μg of protein, determined by the Bradford method [52], with less absorbance indicating stronger labilization of lysosomal membranes.

Hemocyte viability was measured using the trypan blue staining method (modified from Akaishi et al. [53]). In total, 100 μL of hemolymph was centrifuged (500× *g*, for 20 min at 4 °C) to eliminate the humoral compartment, and cells were re-suspended in the same volume of the Alseve medium (Section 5.3). Cell suspension (207,200 ± 30,695 cells mL^−1^) was mixed with 100 µL of the corresponding PST or LC treatment and incubated (Section 5.4.1). In total, 50 μL of trypan blue stain 0.2% in PBS (Sigma-Aldrich, Steinheim, Germany) was added and samples were incubated at 4 °C for 5 min. Live (unstained) and dead cells (stained) were counted within 15 min, using a Fuchs Rosenthal chamber and a light microscope. The number of viable hemocytes mL^−1^ is expressed as percentage of total cells (*n* = 5, each one is an individual mussel).

#### 5.4.3. Cellular Function

Phagocytic activity was assessed using yeast cells stained with Congo red (Sigma-Aldrich, Saint Louis, MO, USA) based on the protocols of Kuchel et al. [54] and detailed in Bianchi et al. [31]. Fifty microliters of hemolymph (207,100 ± 23,057 cells mL^−1^) was mixed with 50 μL of the corresponding PST or LC treatment and incubated (Section 5.4.1). Subsequently, a stained yeast suspension containing twice the amount of yeast cells over the number of viable hemocytes was added to each sample and incubated for 30 min on ice. After the incubation, cells were preserved with 1% glutaraldehyde at 4 °C. Phagocytic activity was calculated by dividing the total number of phagocytosed yeast cells in 100 hemocytes, which were analyzed per sample under the light microscope. Results are expressed as the number of phagocytosed yeast cells per hemocyte (*n* = 5, each one is an individual mussel).

Reactive oxygen species (ROS) production was evaluated in cell suspensions for both intracellular ROS and extracellular H_2_O_2_. After centrifugation of hemolymph (500× *g*, 20 min at 4 °C), the supernatant was removed, and cells were re-suspended in the Alseve medium (Section 5.3) and kept on ice for 30 min. Hemocyte viability was evaluated by the trypan blue stain method (Section 5.4.2) before analysis. Intracellular ROS production was measured using the ROS reactive fluorescent probe dichlorofluorescein-diacetate (H_2_DCF-DA, Invitrogen, Carlsbad, CA, USA) based on the protocol of Moss and Allam [55], with detailed modifications according to Bianchi et al. [31]. Triplicate aliquots of 100 μL containing 50,000 viable cells from each mussel (*n* = 5, each one is an individual mussel) were mixed with 100 μL of the corresponding PST or LC treatment and incubated (Section 5.4.1). After incubation, 40 μL of the supernatant was slowly drawn with a pipette and a final concentration of 40 μM of H_2_DCF-DA was added (200 µL final volume). Changes in fluorescence units were red at excitation/emission: 485/530 nm for 1 h, using a microplate reader (TRISTAR, Berthold, Bad Wildbad, Germany). Results are expressed as fluorescent units (FU) per 10^6^ viable cells. Based on manufacturer’s protocol, extracellular H_2_O_2_ levels were measured using Amplex^®^ UltraRed reagent (AmR, life technologies, Carlsbad, CA, USA), a fluorogenic substrate for horseradish peroxidase (HRP, GE Healthcare, Chicago, IL, USA) that reacts with the hydrogen peroxide (H_2_O_2_) produced and released by cells. Triplicate aliquots of 100 μL containing 50,000 viable cells from each mussel (*n* = 5, each one is an individual mussel) was mixed with 100 μL of the corresponding PST or LC treatment and incubated (Section 5.4.1). After incubation, 40 μL of the supernatant was slowly drawn with a pipette and a final concentration of 50 μM of fluorescent probe and 0.2 U mL^−1^ of enzyme were added (200 µL final volume). Changes in fluorescence units were followed for 1 h at excitation/emission: 550/590 nm, using a microplate reader (TRISTAR, Berthold). Results are expressed as FU per 10^6^ viable cells.

Changes in mitochondrial membrane potential (MMP) were evaluated using the JC-10 fluorescent stain (Enzo Life Sciences ENZ-52305, Doral, FL, USA). Forty microliters of hemolymph (181,500 ± 37,034 cells mL^−1^) was placed on poly-L-lysine coated slides (NeuvitroGG-12-PLL, Vancouver, WA, USA) and incubated for 20 min in a wet chamber to let cells attach to the coverslip. Then, 20 μL of supernatant was slowly drawn with a pipette and 20 μL of the corresponding PST or LC treatment was added. Samples were subsequently incubated for 4 h in the dark. Only the highest doses previously tested for each toxin were applied in this assay (PST: 1250 µM, corresponding to 6900 *A. catenella* cells mL^−1^; LC: culture supernatant from 6800 *A. tamarense* cells mL^−1^). After incubation, the supernatant was gently removed and 40 μL of the JC-10 solution (10 μM, from 1 mM stock in DMSO) was added and allowed to react for 1 h. Fluorescence intensity was measured in 20 adherent cells of each treatment (*n* = 6, each one corresponding to the mean of 20 cells per individual mussel) using a confocal microscope (SP5, Leica, Wetzlar, Germany). Images were analyzed using the software Leica advanced fluorescence lite, and differences in MMP were calculated as the ratio between reversible red fluorescent JC-10 aggregates (excitation/emission: 540/590 nm) and its monomeric, green fluorescent form (excitation/emission: 490/525 nm), and expressed as Reversible:Monomeric (R:M). Autofluorescence in each sample was measured as the control baseline by adding the corresponding DMSO solution in buffer without JC-10.

### 5.5. Scavenger Effects in PST and LC Treatments

#### 5.5.1. In Vitro Chemical Effects of PST and LC Treatments on H_2_O_2_ Levels

Amplex^®^ UltraRed reagent (AmR, life technologies, Carlsbad, CA, USA) is oxidized by H_2_O_2_ in the presence of horseradish peroxidase (HRP, GE Healthcare, Chicago, IL, USA) to produce the red fluorescent compound resorufin (max. 563ex/587em nm). The change of emitted fluorescence intensity is directly proportional to the concentration of H_2_O_2_ in the medium. In preliminary testing, suitable H_2_O_2_ substrate concentrations for this assay were evaluated in a range from 0 to 2 μM (10 μM in deionized water, working solution). H_2_O_2_ was pipetted into a 96-well plate and incubated at room temperature for 2 min with PST 1250 μM (from 6800 *A. catenella* cells mL^−1^) or LC from the 6900 *A. tamarense* cell mL^−1^ culture supernatant (160 μL final volume). Then, final concentrations of 50 μM of fluorescent probe and 0.2 U mL^−1^ of enzyme were added (200 μL final volume). The reagent mix was allowed to react for 5 min in the dark, and final fluorescence was read at excitation/emission: 550/590 nm using a microplate reader (TRISTAR, Berthold, Bad Wildbad, Germany). Data are presented as FU (mean ± standard error of mean) vs. H_2_O_2_ concentrations. Results show a linear scavenging response from both PST and LC (linear regression: Control r^2^ = 0.95; LC r^2^ = 0.83; PST r^2^ = 0.92; *p* ˂ 0.0001 for all) and a significant effect (two-way ANOVA, *p* < 0.0001 for individual effects of algal product and peroxide concentration) on H_2_O_2_ concentrations when used as a substrate at 0.5–1.5 μM (*n* = 4) (Figure 7).

Based on these data, 1.5 μM H_2_O_2_ was chosen as the most sensible concentration to test in vitro scavenging effects of both compound classes. For this, different concentrations of PSTs (250, 500, and 1250 μM, corresponding to 1380, 2760, 6900 *A. catenella* cells mL^−1^) and LCs (culture supernatant from 1700, 3400, and 6800 *A. tamarense* cells mL^−1^) were pipetted in triplicate assays and mixed with H_2_O_2_ into a 96-well plate at a final volume of 160 μL. After 2 min of incubation, AmR and HRP were added and allowed to react for 5 min, as described above. Final fluorescence was read at excitation/emission: 550/590 nm. Results are presented as FU (mean ± standard error of the mean) vs. toxin concentrations (*n* = 4).

### 5.6. Statistical Analysis

Data are presented as mean ± standard error of mean (SEM). Graph Pad Prism 8 (GraphPad Software, Inc., San Diego, CA, USA) and Statistica 7 (StatSoft, Tulsa, OK, USA) were used for data analysis. Normal distribution and homogeneity of variance were checked with the Kolmogorov–Smirnov and Levene’s tests, respectively. Data were transformed to arcsine of the square root when values were expressed as proportions.

Differences among treatments for cytotoxicity and functional variables were assessed using one-way ANOVA of repeated measures and Dunnett’s post hoc comparisons to compare treatments against the control. The dose–response effects between PST/LC concentrations and physiological variables were estimated to fit the appropriate regression models. According to Ritz [56], the exponential model was chosen to describe the toxic and functional effects on hemocytes, since this is the nonlinear model recommended to fit decreasing dose–response patterns that only exhibit asymptotic behavior as the dose (x) range is extensive. Doses were not log-transformed to allow for a more accurate graphical representation of the applied functions. Statistical analyses were applied to the raw data, and an effective concentration 50% (EC_50_) was estimated from this approach. Statistical significance is shown as the standard error of the regression (S). Linear regression was applied to test the scavenging effect of bioactive compounds on ROS (Section 2.2.1 and Section 5.5.1), and statistical significance is shown as the coefficient of determination (r^2^). Two-way ANOVA and Tukey’s post hoc comparisons were used to test differences among groups in the chemical scavenging test, while a one-way ANOVA of repeated measures and Dunnett’s post hoc comparisons were used for the toxin scavenging test on cellular ROS production. One-way ANOVA and Tukey´s post hoc comparisons were applied to assess the toxin effect on MMP. In all cases, differences were considered significant at *p* < 0.05.

The nested two-tailed multiple correlation test was used to assess the relationship among all variables measured in hemocytes for each bioactive compound. Statistical significance was determined by the Pearson correlation coefficient (r^2^) and *p* value < 0.05.

## Figures and Tables

**Figure 1 toxins-13-00544-f001:**
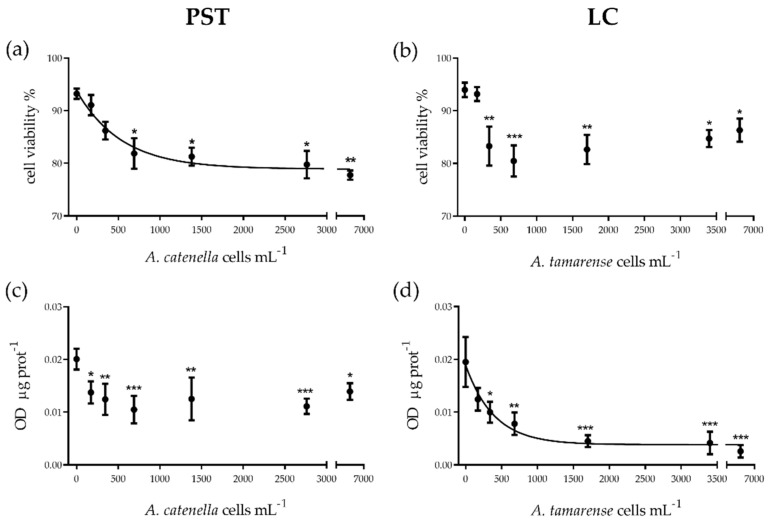
(**a**,**b**) Cellular viability and (**c**,**d**) lysosomal membrane stability of *Mytilus edulis* hemocytes exposed in vitro to paralytic shellfish toxins (PSTs)or to extracellular lytic compounds (LCs) from *Alexandrium* strains (Alex5 and Alex NX-57-08, respectively). PST and LC concentrations are expressed as the equivalent number of *Alexandrium* spp. cells mL^−1^. Results are presented as mean ± SEM. Asterisks denote significant differences between an individual treatment and the control (* *p* ˂ 0.05; ** *p* ˂ 0.01; *** *p* ˂ 0.001), *n* = 4 (**c**,**d**), pooled hemolymph from two mussels and *n* = 5 (**a**,**b**), individual mussels per treatment.

**Figure 2 toxins-13-00544-f002:**
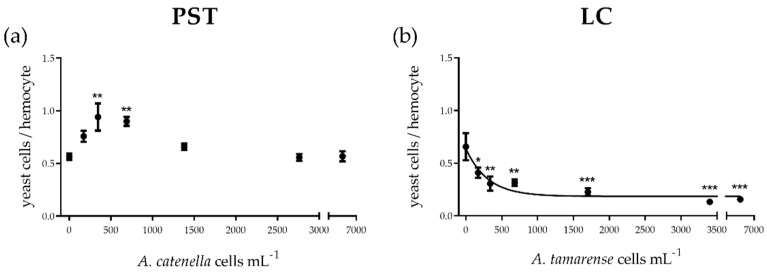
Phagocytic activity of *Mytilus edulis* hemocytes exposed in vitro to (**a**) PSTs or to (**b**) extracellular LCs from *Alexandrium* strains (Alex5 and Alex NX-57-08, respectively). PST and LC concentrations are expressed as the equivalent number of *Alexandrium* spp. cells mL^−1^. Results are presented as mean ± SEM. Asterisks denote significant differences between an individual treatment and the control (* *p* ˂ 0.05; ** *p* ˂ 0.01; *** *p* ˂ 0.001), *n* = 5 mussels per treatment.

**Figure 3 toxins-13-00544-f003:**
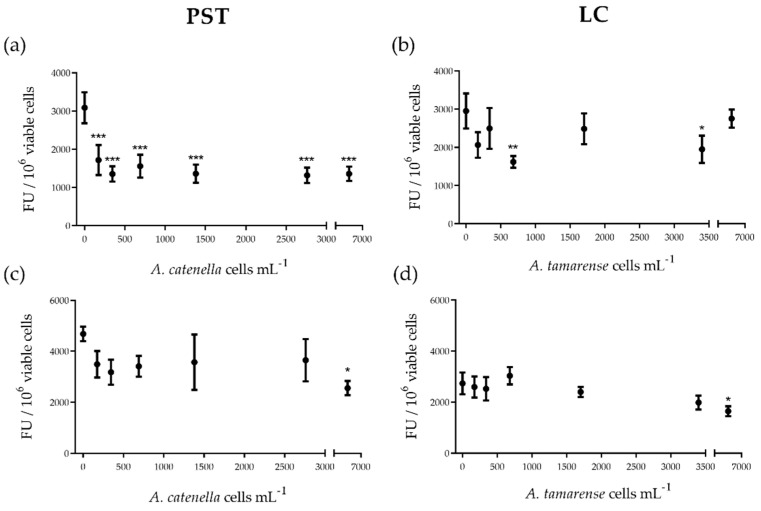
(**a**,**b**) Extracellular H_2_O_2_ and (**c**,**d**) intracellular ROS levels in *Mytilus edulis* hemocytes exposed in vitro to PSTs or to extracellular LCs from *Alexandrium* strains (Alex5 and Alex NX-57-08, respectively). PST and LC concentrations are expressed as the equivalent number of *Alexandrium* spp. cells mL^−1^. Results are presented as mean ± SEM. Asterisks denote significant differences between an individual treatment and the control (* *p* ˂ 0.05; ** *p* ˂ 0.01; *** *p* ˂ 0.001), *n* = 5 mussels per treatment.

**Figure 4 toxins-13-00544-f004:**
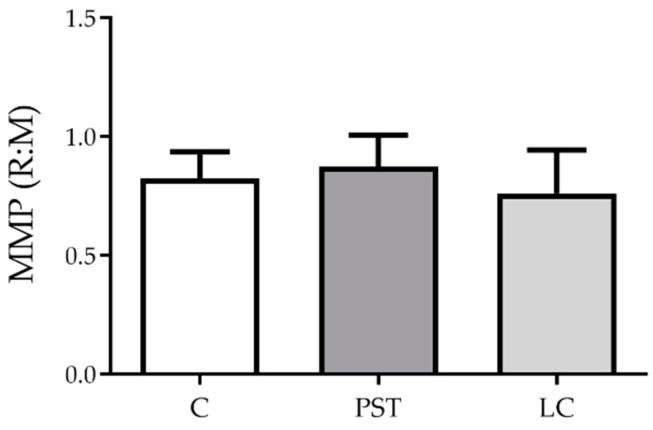
Equivalent to the mitochondrial membrane potential (MMP) given as the ratio of reversible red staining and mononomeric green staining R:M in *Mytilus edulis* hemocytes exposed in vitro to PSTs and extracellular LCs from *Alexandrium* strains (Alex5 and Alex NX-57-08, respectively). JC-10 stain of control (C), PST (1250 µM, corresponding to 6900 *A. catenella* cells mL^−1^) and LC (supernatant from 6800 *A. tamarense* cells mL^−1^) treatments. Results are presented as mean ± SEM, *n* = 6 (mussels per treatment).

**Figure 5 toxins-13-00544-f005:**
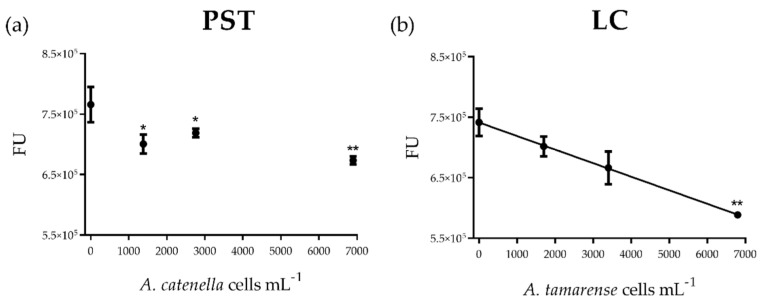
In vitro scavenger effect of (**a**) PSTs and (**b**) LCs from *Alexandrium* strains (Alex5 and Alex NX-57-08, respectively) on H_2_O_2_. PST and LC concentrations are expressed as the equivalent number of *Alexandrium* spp. producer cells mL^−1^. Results are presented as mean ± SEM. Asterisks denote significant differences between an individual treatment and the control (* *p* ˂ 0.05; ** *p* ˂ 0.01), *n* = 4 per treatment.

**Figure 6 toxins-13-00544-f006:**
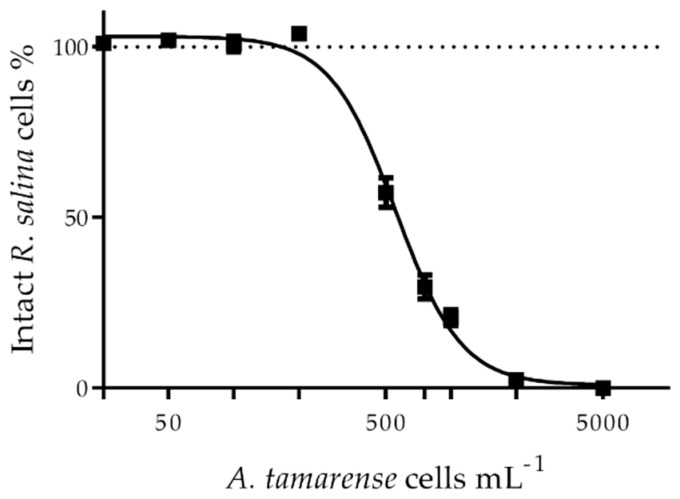
*Rhodomonas salina* bioassay showing lytic activity in the culture supernatant of *Alexandrium tamarense* (AlexNX-56-08), used for the in vitro exposures of *Mytilus edulis* hemocytes. Results are presented as mean ± SD, *n* = 3.

**Figure 7 toxins-13-00544-f007:**
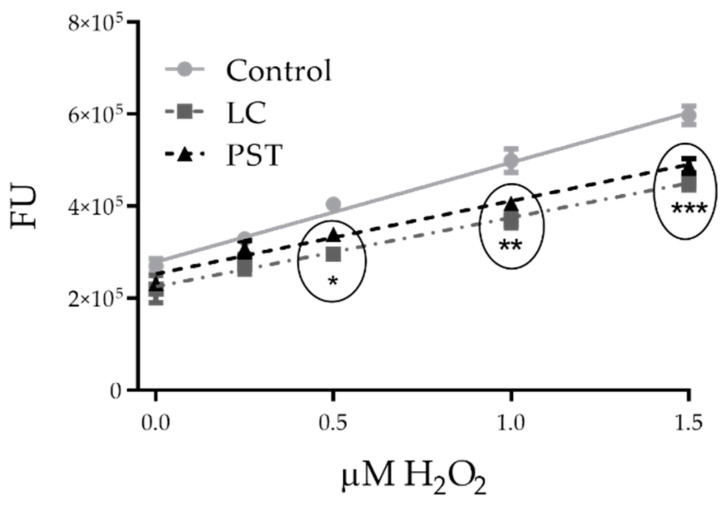
Preliminary testing of H_2_O_2_ concentrations as substrate for Amplex^®^ UltraRed in a toxin scavenging capacity assay (PST: 1250 µM, corresponding to 6900 *Alexandrium catenella* cells mL^−1^; LC: culture supernatant from 6800 *Alexandrium tamarense* cells mL^−1^). Results are presented as mean ± SEM. Asterisks denote significant differences among treatments for a given concentration (* *p* < 0.05; ** *p* < 0.01; *** *p* < 0.001), *n* = 4 per treatment.

**Table 1 toxins-13-00544-t001:** Pearson correlation matrix (coefficient of correlation, r^2^) for cytotoxic and functional variables tested in *Mytilus edulis* hemocytes exposed in vitro to paralytic shellfish toxins (PSTs) or to extracellular lytic compounds (LCs) from *Alexandrium* strains (Alex5 and Alex NX-57-08, respectively).

	**PST**	**1**	**2**	**3**	**4**	**5**
1	Cell Viability	1.00				
2	LMS	0.68 *	1.00			
3	Phagocytosis	0.14	−0.46	1.00		
4	Intracellular ROS	0.68 *	0.65	−0.25	1.00	
5	Extracellular H_2_O_2_	0.76 *	0.91 **	−0.30	0.82 *	1.00
	**LC**	**1**	**2**	**3**	**4**	**5**
1	Cell Viability	1.00				
2	LMS	0.71 *	1.00			
3	Phagocytosis	0.69	0.98 ***	1.00		
4	Intracellular ROS	0.03	0.63	0.65	1.00	
5	Extracellular H_2_O_2_	0.45	0.30	0.34	−0.33	1.00

* *p* ˂ 0.05; ** *p* ˂ 0.01; *** *p* ˂ 0.0001. LMS: lysosomal membrane stability.

**Table 2 toxins-13-00544-t002:** Stock PST concentrations (ng µL^−1^ and µM) measured in acetic acid extracts of *Alexandrium catenella* cells (Alex5 strain) and used for the in vitro exposures of *Mytilus edulis* hemocytes.

PST	ng µL^−1^	µM
STX	4.3	14
NEO	63.5	200
GTX 1/4	105.2	255
GTX 2/3	0.7	2
C1/C2	94.3	199
Total concentration	268.1	670

**Table 3 toxins-13-00544-t003:** Estimated number of *Alexandrium* spp. cells mL^−1^ (Alex5 and Alex NX-57-08) corresponding to PST (µM) and LC (% of culture supernatant) concentrations used for the in vitro exposures of *Mytilus edulis* hemocytes.

	µM/%	Cells mL^−1^
PST	1250	6900
	500	2760
	250	1380
	125	690
	62.50	345
	31.25	172.5
LC	50	6800
	25	3400
	12.5	1700
	5	680
	2.5	340
	1.25	170

## Data Availability

The data presented in this study are openly available on Zenodo (https://zenodo.org/, accessed on 7 June 2021) at doi:10.5281/zenodo.4906093.

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
