# Peer review of "In Vitro Effects of Paralytic Shellfish Toxins and Lytic Extracellular Compounds Produced by Alexandrium Strains on Hemocyte Integrity and Function in Mytilus edulis"

_toxins, 2021, doi:10.3390/toxins13080544_

Round 1

Reviewer 1 Report

This manuscript presents an in vitro experiment testing the effect of two types of compounds, each produced by two strains of Alexandrium spp. on hemocyte variables of mussels.   The experiments are well conducted, the methods are sound, and the results are adequately presented and discussed. One originality of this study is that it compares in the Discussion in vitro effects identified from the present work with in vivo results acquired in a previous study. The manuscript is well written, although some details appear somewhat unclear, particularly in the Material and Methods (see specific comments below). Overall, it is a useful study that contributes to highlight the importance of considering BEC/LC produced by Alexandrium spp. when assessing their toxicity to marine fauna, and not only PST as it is unfortunately often still the case in many papers.

Graphical abstract:

Both PST and LC induced similar decrease in viability, which should appear in the graphical abstract. From this graphical abstract, it seems that PST are not/less cytotoxic than LC, which is actually not evidenced by the hemocyte viability results. However, what is clear is that the mechanisms of action are different.

Abstract

- Line 9-10: Here and generally elsewhere in the manuscript: please clarify that the results of this study are specific to the Alexandrium strains you have tested, since effects depend on LC and PST composition and concentrations, which are highly strain-dependent, and these results cannot be inferred to the species level unless many strains of each species would have been tested.

- L.10-11: the duration of the exposure could be mentioned in the abstract.

Introduction:

Additional bibliographic references could be of interest in your Introduction:

- A recent review of the effects of HABs on bivalve hemocytes, which also discuss the effects of BECs/LCs vs PST:

Lassudrie, M., Hégaret, H., Wikfors, G.H., Mirella da Silva, P., 2020. Effects of marine harmful algal blooms on bivalve cellular immunity and infectious diseases: A review. Dev. Comp. Immunol. 108, 103660. https://doi.org/10.1016/j.dci.2020.103660

Mardones, J.I., Dorantes-Aranda, J.J., Nichols, P.D., Hallegraeff, G.M., 2015. Fish gill damage by the dinoflagellate Alexandrium catenella from Chilean fjords: Synergistic action of ROS and PUFA. Harmful Algae 49, 40–49. https://doi.org/10.1016/j.hal.2015.09.001

- L.84-85: “the contrary is true”: this indirect formulation is somewhat confusing.

- L.101-109: this part does not belong to the Introduction, it is a summary of the results, already mentioned in the Abstract.

Results

- L113-122: Results of hemocyte concentrations are not presented, although they are linked to the viability results: some hemocytes may not be counted as dead if they are too lysed to be detected, but the total hemocyte count would then decrease.

-L114: Here and elsewhere: The most usual dose-response models in ecotoxicology are log-logistic models or Weibull models – were those models tested before the exponential models selected here? For more precision on the methodology, see:

Ritz, C., 2010. Toward a unified approach to dose-response modeling in ecotoxicology. Environ. Toxicol. Chem. 29, 220–229. https://doi.org/10.1002/etc.7

Ritz, C., Baty, F., Streibig, J.C., Gerhard, D., 2015. Dose-response analysis using R. PLoS One 10, 1–13. https://doi.org/10.1371/journal.pone.0146021

-L117: Here and elsewhere: the results of the ANOVA test should be provided before the results of the posthoc test.

- L125: What does “LMS” stand for?

- L 127: did you mean “p<0.05” ?

- L151 and Figure 3a,b: “extracellular ROS”: it would be more adequate to specify “extracellular H2O2”, considering the method used.

- L151-152: is it strictly the “production” of H2O2 that is impaired, or could the LCs also have scavenged the H2O2 released in the hemocyte medium just after being excreted? From the results of your H2O2 scavenging experiment, I think both process could be involved, as you also aknowledge L245 in the Discussion.

Figures:

- Have you tried to use a log scale to avoid cutting the x axis?

- Fig. 3c,d: It is surprizing that the intracellular ROS results in the controls are so different between PST and LC; were the hemocytes used in the same conditions?

- Fig. 4: Editing of the x-axis is not homogenous compared with the other figures, and the 0 value is missing.

-Fig.5a: Units: “A:M” in the graph vs “R:M” in the legend.

- Fig.5b: I do not think it is useful to keep these photographs in the results since you show there are no differences between conditions, especially since the pictures chosen do not seem to be representative of the data shown above in Fig.5a : LC staining is more intense on the pictures than the other conditions. Also, interestingly, the distribution of the staining within the cell seems different between conditions: is it a feature that you were able to analyse? 

Figure captions:

- In all figure legends: please mention that data are presented as “mean ± SE”

- “Alexandrium spp;” : “spp.” should not be in tialics

- “ * p<0.5 ” : I believe you meant “* p<0.05”

- The number and the nature of the replicates presented is unclear and present discrepancies with the Material and Methods section: for example, lysosomal membrane stability was analysed using duplicate aliquots of a pool of hemolymph from 2 mussels, according to the Material and Methods L.437, whereas captions from Figure 1 mentions n=4-5 per treatment.   Please always clarify what represent each replicate (different pools of hemolymph? Different mussel hemolymph samples? etc).

Discussion:

-L.200:  “not lytic bacteria may be present”: unclear, do you mean “bacteria (lytic and non lytic) may be present”

- L.214: “digestive enzymes”: rather refer to food digestion, “intracellular lytic enzymes” may be more adequate

-L219: “on the contrary” is confusing here as it does not refer to the last sentence.

- L225: reference missing to evidence the presence of sterols and phosphatidylcholine in mussels/bivalve hemocytes.

- L236: “PST harmful effect…are not strong enough to impair phagocytosis” : despite the “immunostimulant” effects of PST extracts on live hemocytes as indicated by phagocytosis activity results, it should not be forgotten that PST extracts induced significant hemocyte mortality, which would impair global immune capacity. 

- L243: “ROS production does not correlate to the phagocytic activity”: from the Material and Methods, I do not understand that you tested the correlations between those variables?  It would be interesting to run a multiple correlation test on you whole dataset to identify the links between the effects observed (e.g. responses on viability, lysosomal membrane stability and phagocytosis seem to be linked), and to better integrate the hemocyte responses.

- L256: again, a correlation test would formalize the links that you discuss between phagocytosis and subcellular variables.

- L266 “lipid peroxidation”.. see also Mardones et al 2015 previously mentioned who discuss the involvement between ROS, fatty acids and toxins in ichthyotoxicity

- L277: exposure to “PST and LC producer algae”: at similar algal density as here?

- L279: comparison of in vivo and in vitro results: to what extent were the exposed mussels of the 2 different studies comparable (maintaining conditions, physiological state (reproduction), size/age, orgine, etc)?

- L310: Did you detect tissue damage in this previous study?

- L311-315: This is even more true when considering the many different functions hemocytes are involved in, and which can be affected by HAB exposure (immunity, tissue repair, nutrition, biomineralization..), inducing indirect responses from hemocytes (see also Lassudrie et al 2020 previously mentioned).

- L318: please specify that the “immunostimulant” effect is observed only at low concentrations

-L319: … but causing similar hemocyte mortality than LC! Although more effects were observed at subcellular level after LC exposure (highlighting that different toxicity mechanisms are involved) both compound types induced similar mortality levels, which should not be minimized.

- L326-328: The conclusion could include a word about how realistic the range of algal densities chosen are compared to the field, ie. how the effects observed in the in vivo and in vitro experiments can be applied in the field.

Material and Methods

- L341: Could you please specify the algal densities at time of harvest? And were the strains cultivated in plastic or glass flask, and in what volume?

-L344: “were transferRED”

- L347: “were filterED”

- L353: Wrong references, you probably meant 41,42

- L361: “whole cell cryptophyte Rhodomonas salina … bioassay”: can you clarify whether “whole cells” applies here to Alexandrium?

- L362: Please specify what type of non-linear regression was fitted (see previous comment L 114)

- L366: centrifugation at 3200xg 15 min seems quite strong, it may have induced LC release into the medium. How did you select these centrifugation parameters, did you check the effect of centrifugation speed and time on LC amount (as quantified by toxicity to Rodomonas for example?)

- L381-382: can you add some precisions: Was the seawater filtered, and at what pore size (could the water contain natural phytoplankton?)? How many mussels were maintained per volume of water, and what was the renewal rate of the tanks?

- L383: This feeding frequency is low, usually bivalves maintained for such experiments are fed at least once a day. Did you observe any mortality during this acclimatation period?

- L405: “p<0.05” means there was actually a significant difference of viability in time. However, a viability dropping from 96% to 93% seems really acceptable for such experiments.

- L421: “density” or “concentration” instead of “amount” of A. tamarense cells

- Table 2: The algal strains could be specified again here as those information are really specific to the strains used

- L430-433: Here, from my understanding, 1 replicate = 1 hemolymph sample from one mussel, with 6 replicates total, ie. 6 mussels. However, this paragraph does not seem to be applicable to all assays (L 347 for example), it might be clearer to specify in each assay section the number and nature of replicate used.

- L 347: From my understanding, lysosomal membrane stability assay was conducted using 2 replicates, with one replicate = one pool of hemolymph from 2 mussels. Did the 2 replicates correspond to 2 different hemolymph pools then or was the same pool tested twice?

- L444: mention the method used for protein quantification

-L466: From this first sentence, it could be understood that ROS was determined immediately after centrifugation, please re-formulate.

- L472 and 482: Three replicates for each of the 6 mussel hemolymph samples mentioned in sectioned 5.4.1? Unclear.

- L473 and 483: Was this ratio of number of hemocytes : volume of LC or PST, always comparable between the different assays (Viability, ROS, phagocytosis, etc)?  Please mention the hemocyte concentrations used in the other assays.

- L482: please clarify whether the hemolymph samples were first centrifuged and hemocytes resuspended, as for intracellular ROS. Also, “50,000” or “50 000” if you mean fifty thousands and not fifty.

L 490: Both “ ROS” and “ generation” seems inadequate, since “ROS” are in fact specifically “H2O2”, and H2O2 is not generated by any cells in this experiment which consists in adding H2O2 in the media exempt of hemocytes.

L 521: This section on Hemocyte membrane potential would better fit in the section “Hemocyte functions” than “Scavenging effects”

L532: As for the other assays, the nature of the replicate is unclear (6 mussel hemolymph samples?)

L545-547: see previous comments regarding dose-response models

L556: it is the opposite, significant differences are considered significant when p <0.05 and not p>0.05 !

Author Response

We have included the detailed and useful suggestions of the reviewer N°1, and we feel that the manuscript is much improved thanks to these inputs.

Reviewer 2 Report

Dear authors,

This paper presents an important advance towards a better understanding of the impacts PSTs have on marine biota. I would like to start by saying that this paper was easy to read and digest and well written. I appreciated the opening comments on the discussion regarding the possible interpretation of the data and comparison with other studies. It was also noted the complexity of the methodology and it was very well detailed. The preliminary tests (figs. 6 and 7) were also a very good addition.

Section 3.2 could benefit from a more thorough discussion, it seems like you're going back and forth with only your 2 papers. Despite having few studies to compare your results, there are some, either in vivo or in vitro, separately or in other bivalve species. In the absence of comparable metrics, you can discuss overall results/effects of both approaches in bivalves. For me, this somewhat less strong section is the greatest weakness of the present work.

I will now refer more specific comments:

  1. Between line 89-98 and 101-109: this is discussion material, please remove and elaborate a bit on the objectives. This work deserves a more detailed description of the objectives.
  2. Line 138: "respect" should be "in respect to"
  3. Line 200: "compounds, and not lytic bacteria may also be present, as algae cultures are not axenic." This sentence is not completely clear, please rephrase.
  4. Line 261: "evidence" should be"evidences".
  5. Line 263: "microzooplancton" is with K and not ".
  6. Line 264: Genus name is not in italics.
  7. Line 344: "transfer" should be"transferred".
  8. Line 346: "to be homogenize" should be "to be homogenized".
  9. Line 347: "filter through out" should be "filtered through".
  10. Line 482: "50.000", previously you used 50,000, please standardize.
  11. Line 501: "5 min at dark" should be "in the dark".

Author Response

We have included the detailed and useful suggestions of the reviewer N°2, and we feel that the manuscript is much improved thanks to these inputs.
